# Modeling and Performance Optimization of Unmanned Aerial Vehicle Channels in Urban Emergency Management

Bing Han [1], Danyang Qin [1,*], Ping Zheng [1], Lin Ma [2] and Merhawit Berhane Teklu [3]

1   Department of Electronic and Communication Engineering, Heilongjiang University, Harbin 150080, China; 2201701@s.hlju.edu.cn (B.H.); 2201700@s.hlju.edu.cn (P.Z.)
2   Communication Research Center, Harbin Institute of Technology, Harbin 150001, China; malin@hit.edu.cn
3   School of Electrical and Computer Engineering, Dire-Dawa University, Dire Dawa 1362, Ethiopia; merhawit.birhane@ddu.edu.et
*   Correspondence: qindanyang@hlju.edu.cn

**Abstract:** With the development of smart cities, the use of unmanned aerial vehicles (UAVs) for interactive information exchange between air and ground can provide effective support for the deployment of emergency work. However, the existing UAV air-to-ground channels often use a single channel model. Considering that the density and distribution of obstructions on information transmission paths at different heights are different, only using a single channel model greatly affects the reliability of communications. Aiming at addressing the different channel characteristics of air-to-ground channels at different heights, a height-based adaptive SUUL-SULA channel model is proposed in this paper. Firstly, in the ultra-low altitude environment, the influence of large-scale fading and small-scale fading on the envelope of the received signal is discussed based on the classic LOO model, and the probability density function and bit error rate model of the received signal are derived. Secondly, a SULA channel model based on Jakes' model is proposed in the low-altitude environment. The uniform circular array beamforming technology is adopted to realize the design of the Doppler frequency shift compensation algorithm. Finally, the simulation results show that the SUUL-SULA model effectively reduces the bit error rate of the system and improves the reliability of communication. Therefore, this model can provide effective physical support for the application of UAV in smart city emergency management.

**Keywords:** emergency management; channel model; shadow fading; beamforming; Doppler effect

## 1. Introduction

With the continuous integration of information technology and traditional working modes, the application of Unmanned Aerial Vehicles (UAVs) in urban emergency management has attracted more and more attention. The UAV can be used as a mobile sensor platform to realize information interaction between air and ground. The UAV can monitor a scene in real time and transmit data information to emergency managers. This assists the emergency managers in accurately knowing the situation on the spot, saving lives and controlling the crisis in time. At present, most studies on air-to-ground channel models are focused on a single channel model without considering the height of the UAV. However, in a complex urban environment, the distribution density and location of complex scatterers such as buildings and trees vary with height. They make the signal experience different loss and attenuation when transmitted at different heights. Therefore, there are actually huge differences in channel characteristics at different heights. In this case, the single-channel model has some limitations, especially in emergency management, where communication reliability is highly required. Using only a single channel model will make the information unstable during transmission, which causes several problems such as interference and packet loss. Therefore, it is very necessary to establish an adaptive and more accurate

channel model in an urban environment to ensure the reliability of information transmission in emergency situations. As a result, the impact and harm caused by emergencies on individuals and society will be reduced.

Consequently, we propose to divide the UAV communication area into two parts: ultra-low altitude (0~100 m) and low altitude (100~1000 m). Firstly, the urban channel characteristics are analyzed. The analysis of urban channel characteristics is shown in Figure 1. In the ultra-low altitude area, reflection, refraction, and scattering caused by a variety of complex scatterers during signal transmission are considered by the SUUL channel model. The model focuses on shadow fading and multipath fading. In addition, the probability density function of related parameters and bit error rate are derived based on the improved LOO model. In the low-altitude area, the influence of shadow fading on the channel gradually decreases as the flying height of the UAV increases. The Doppler frequency shift spread caused by the flying speed of the UAV leads to severe channel fading. As a result, the SULA channel model proposed in this paper adopts the uniform circular array beamforming technology to design the Doppler frequency shift compensation algorithm. Finally, the UAV adjusts the channel model adaptively according to the height. The reliability of UAV communication is guaranteed and the effective monitoring range of UAVs has also been improved.

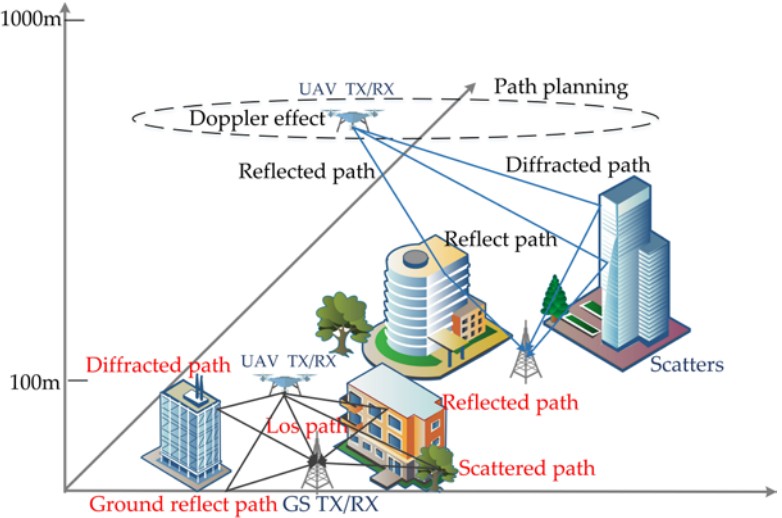

**Figure 1.** The analysis of channel characteristics.

## 2. Related Work

Traditional wireless channel modeling methods are mainly divided into deterministic modeling methods and statistical modeling methods. Deterministic channel modeling analyzes the fading mechanism of received signals based on electromagnetic wave transmission theory. Maxwell's equations, scatterers, environmental topography and other information are adopted by this method. In [1], Feng et al. derived path loss and shadow models from simulated propagation data extracted from outdoor deterministic ray tracing models. Based on the statistical parameters of the urban environment in the target area, Al-Hourani et al. proposed a radio frequency propagation model that predicted the path loss between the low-altitude platform and the ground receiver [2]. However, the accuracy of the deterministic modeling method depends on the database of the modeling environment, and it requires a lot of complex calculations. Thus this modeling method has certain limitations in the application scenario of emergency management. In addition, the Doppler frequency shift caused by the high-speed movement of the UAV was not considered in the above model building process.

Statistical modeling methods can be divided into channel impulse response modeling and stochastic channel modeling. Channel impulse response modeling uses measurement equipment to measure and store the channel impulse response in real time. The stored

data can be reused, and a simple mathematical model can be established based on the measured data. The models established by this method are mostly tapped delay line models. For example, Zaman et al. used a tapped delay line filter model with time-varying coefficients to model the channel [3]. It is assumed that the amplitude attenuation of the NLOS component obeys the Rayleigh distribution, and the multipath fading obeys the Gaussian distribution. Correspondingly, the probability density function is used by stochastic channel modeling to describe the channel fading parameters, and then the relevant parameters are adopted to establish a channel model of a similar communication environment. The classic stochastic channel model includes Rayleigh fading channel model [4], Rice fading model [5], lognormal fading model [6], Suzuki model [7], and Clarke model [8] and other classic models. In addition, the stochastic model is helpful to analyze the time-varying characteristics of UAV channels [9]. The stochastic channel modeling method is suitable for designing various wireless communication systems. Statistical theory is adopted by this method to establish a reasonable channel model, and the physical meaning is relatively intuitive. Therefore, considering the channel characteristics and modeling complexity of the complex urban environment, the random channel modeling method is selected for follow-up research.

In addition, in the past research on channel models, the research on typical ground-to-ground communications has been very mature. Cotton proposed a shadow fading model [10]. The statistics of the received signal are represented by the clustering of multipath components. However, the channel characteristics of the air-to-ground channel are different from that of the terrestrial communication channel. Compared with terrestrial communications, UAVs are in a complex and changeable wireless environment, and their BER requirements are much higher than those of traditional terrestrial wireless channels. The study of Doppler frequency shift which is a key parameter needs to be added to the calculation of channel parameters due to the high maneuverability of UAVs.

On the other hand scholars have proposed improvements to the classic LOO model [11]. Simunek et al. researched on low-altitude channels in urban areas based on the LOO model, and also developed a narrowband time series generator [12]. However, this study did not pay attention to the important position of communication reliability in channel performance. In addition, in the above studies, a single channel model is used to characterize the channel. In fact, in a complex urban environment, due to the different heights of scatterers such as buildings and trees and changes in their distribution positions, channel characteristics at different heights are different. The single channel model has insufficient communication reliability in this case.

Correspondingly, some scholars also proposed a hierarchical adaptive channel model. Yang et al. proposed an adaptive channel based on the improved LOO model and the Rice model [13], in which the LOS component obeyed the lognormal distribution and the multipath component obeyed the Rayleigh distribution. Directional antennas were used for communication by the improved Rice model. This article successfully expanded the degrees of freedom of the two models. However, communication reliability has not been fully discussed. In addition, directional antennas have poor performance in high mobility environments. The omnidirectional antenna has the advantages of large coverage, low price, and better performance in high mobility scenarios. Therefore, in the urban emergency management environment of UAVs, considering the use of omnidirectional antennas will have better performance.

Based on the discussion of the above research, an adaptive SUUL-SULA channel model is proposed in this paper. The channel characteristics are described more accurately through this model. UAVs could select better channels according to different flight altitudes to improve communication reliability to a greater extent. It shows better performance in urban emergency management. The specific roadmap of the research is shown in Figure 2.

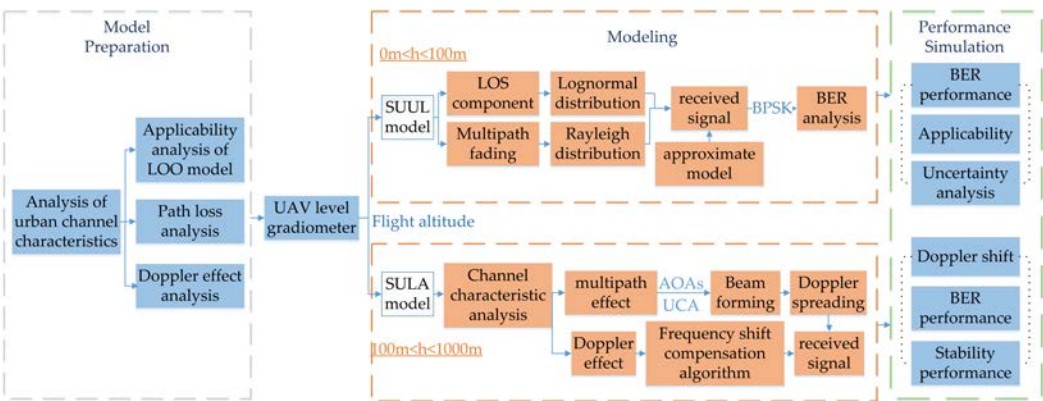

**Figure 2.** Roadmap of the research.

Main contributions of this paper are as follows:

(1) A height-based adaptive air-to-ground channel is proposed based on the channel characteristics of different heights in the urban environment to improve channel reliability;

(2) In the ultra-low altitude area, BPSK method is adopted to model the ultra-low altitude communication channel of UAV with reference to LOO model. In the low altitude region, the separation and compensation of Doppler frequency shift are emphasized by using beamforming technology and multipath characteristics;

(3) The simulation experiment proves that this model can effectively reduce the bit error rate of communication and improve the reliability of communication. It provides effective physical level support for the application of UAV in smart city emergency management.

## 3. Analysis of Channel

In a complex urban environment, the distribution density of buildings and vegetation varies with height. This leads to different main factors for signal fading at different heights, and channel characteristics are more complex and changeable. The fading characteristics of wireless communication channels also have great randomness. In addition to direct radiation, when radio waves encounter obstacles, they will reflect, diffract, and scatter according to their wavelength and the size of the obstacle. This leads to a certain amount of radio wave energy loss and cause signal fading. This section gives a brief overview of the fading model that will be used.

### 3.1. Loo Model

In 1985, Loo proposed the so-called Loo model for statistical channel of land mobile satellite link [14]. The model assumes that the received signal is composed of a LOS component and a multipath component. The LOS component obeys the Lognormal distribution and the multipath component obeys the Rayleigh distribution. That is, the received signal is:

$$r(t) = z(t)s(t) + d(t) \tag{1}$$

where $z(t)$ is the LOS component, $s(t)$ is the shadow fading, and $d(t)$ is the multipath component.

Assuming that the amplitude $z$ of the LOS component remains constant, the received signal envelope $r$ obeys the Rice distribution:

$$f_r(r|z) = \frac{r}{\lambda_0^2} \exp\left(-\frac{r^2 + z^2}{2\lambda_0^2}\right) I_0\left(\frac{rz}{\lambda_0^2}\right) \tag{2}$$

where $\lambda_0^2$ is the average scattered multipath power, $I_0(\cdot)$ is the first kind of zeroth order modified Bessel function.

$z(t)$ obeys Lognormal distribution:

$$f(z) = \frac{1}{z\sqrt{2\pi d_0}} \exp\left(-\frac{(\ln z - \mu)^2}{2d_0}\right) \tag{3}$$

where $\mu$ is the mean deviation of $\ln z$, $d_0$ is the variance of $\ln z$.

According to the total probability formula, the probability distribution of the received signal is shown in Equation (4):

$$f(r) = \int_0^\infty f_r(r|z) f(z) \mathrm{d}z = \frac{r}{\lambda_0^2 \sqrt{2\pi d_0}} \int_0^\infty \frac{1}{z} \exp\left(-\frac{r^2 + z^2}{2\lambda_0^2} - \frac{(\ln z - \mu)^2}{2d_0}\right) \mathrm{d}z \tag{4}$$

when the signal amplitude is much larger than $\lambda_0$, $f(r)$ obeys the Lognormal distribution:

$$f(r) = \frac{1}{r\sqrt{2\pi d_0}} \exp\left(-\frac{(\ln r - \mu)^2}{2d_0}\right) \tag{5}$$

When the signal amplitude is much smaller than $\lambda_0$, $f(r)$ obeys the Rayleigh distribution:

$$f(r) = \frac{r}{\lambda_0^2} \exp\left(-\frac{r^2}{2\lambda_0^2}\right) \tag{6}$$

The residual cumulative probability distribution function of $r$ is:

$$C_R(r) = P(r > R) = \int_R^\infty f(r) \mathrm{d}r = 1 - \int_0^R f(r) \mathrm{d}r \tag{7}$$

where R is a certain value.

The Loo model used a helicopter to simulate satellites and flied at an elevation angle of 15° relative to the receiver in light shadowing and heavy shadowing areas where the tree coverage was about 35%. Model parameters are shown in Table 1 [15].

**Table 1.** Parameters of Loo model.

| Conditions | Light Shadowing | Heavy Shadowing |
|---|---|---|
| $10\lg\left(\sqrt{d_0}\right)$ | 0.5 | 3.5 |
| $10\lg(\mu)$ | 0.5 | −17 |
| $10\lg\left(\lambda_0^2\right)$ | −8 | −12 |

Figure 3 is the fitting diagram of cumulative distribution curve of the theoretical data and the measured data. Figure 3a is the cumulative distribution curve of the Loo model theoretical data and the measured data under the light shadowing, while Figure 3b shows the cumulative distribution curve under heavy shadowing conditions.

In Figure 3, it can be observed that the received signal envelope and the LOS component are affected by the shadowing. Moreover, with the increase of cumulative probability distribution, the theoretical and measured values of the received signal envelope decrease non-linearly. In a light shadowing environment, the fitting effect between the calculation results and the measured values is average. The calculation results reflect a higher shadowing effect. In contrast, the model has a better fitting effect in heavy shadowing environments. In addition, studies have shown that land mobile satellite systems are similar to urban UAV air-to-ground links. According to the analysis of statistical data and measured data, the LOO model is a suitable starting point for improving the model [12]. Therefore, the LOO model has been improved for urban ultra-low altitude areas with heavy shadow fading.

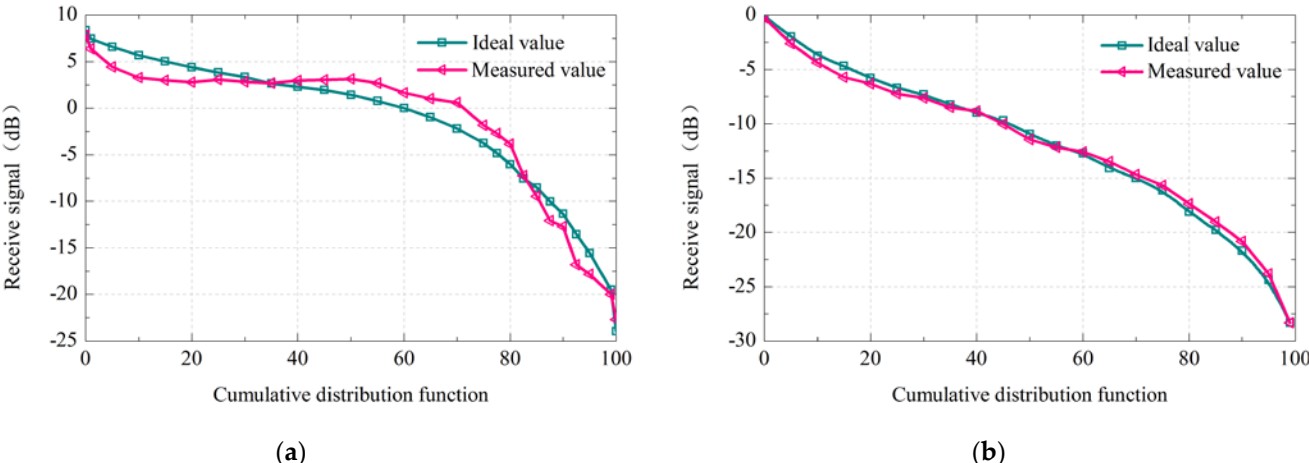

**Figure 3.** A comparison of measured and calculated values of the signal of Loo model. (**a**) The light shadowing environment; (**b**) The heavy shadowing environment.

### 3.2. Large-Scale Path Loss Model

Large-scale fading is represented by path loss and shadow fading. Path loss increases with the increase of communication distance. Shadow fading varies along the path loss. In the emergency management of the complex urban environment, there are some obstacles that cause shadow fading, such as buildings, trees, and others. When the receiving end moves to the shaded area, the received signal strength decreases. And the obstructions with different distribution densities and positions are distributed on different transmission paths. Even if the communication distance is the same, the received signal strength is different.

The free space propagation model is the simplest path loss model, which is used to predict the power of received signal in wireless communication environment. The received signal power can be expressed by the Friis equation as Equation (8):

$$P_r = P_t G_t G_r \left(\frac{\lambda}{4\pi d}\right)^2 \tag{8}$$

where $P_r$ is the received signal power, $P_t$ is the transmit signal power, $G_t$ is the transmit antenna gain, $G_r$ is the receive antenna gain, $\lambda$ is the wavelength and meets $\lambda = c/f$, $d$ is the distance of radio wave transmission.

When there are no obstacles in propagation path, it is approximated as free space propagation. Path loss is expressed as:

$$PL = 10\lg\frac{P_t}{P_r} = 20\lg\left(\frac{4\pi fd}{c}\right) \tag{9}$$

where $f$ is the signal transmission frequency.

The logarithmic distance path loss model is established by introducing the path loss index $\alpha$ that related to the environment. The logarithmic distance path loss model is expressed as:

$$PL = PL_{ref} + 10\alpha\lg\left(\frac{d}{d_{ref}}\right) \tag{10}$$

where $d_{ref}$ is the reference distance, $\alpha$ is the path loss index. Different path loss indexes correspond to different environments.

In general, $PL_{ref}$ represents the path loss gain in free space at distance $d_{ref}$. When $G_t = G_r = 1$, the transceiver antenna is unity gain, and the average received signal power is expressed as:

$$\overline{S} = P_t \left( \frac{\lambda}{4\pi d_{ref}} \right)^2 \left( \frac{d_{ref}}{d} \right)^{\alpha} \tag{11}$$

As the UAV keeps moving during the communication process, there are a large number of different obstructions on different transmission paths. Even if the communication distance between the sender and receiver is same, the signals propagating along different paths will have different path losses. The lognormal shadow model is introduced in consideration of the above practical problems. The lognormal shadow model is expressed as:

$$PL = PL_{ref} + 10\alpha \lg \left( \frac{d}{d_{ref}} \right) + X \tag{12}$$

where $X$ is a random variable with Zero-mean Gaussian distribution. The variance of $X$ is $\sigma_X^2$, that is, $X \sim N(0, \sigma_X^2)$. The shadow variance can be estimated in real time based on empirical measurements [16]. When the communication distance is the same, the model can describe the actual shadow effect, and can reduce the error between the actual value and the estimated value of the path loss.

### 3.3. Doppler Effect

The Doppler effect can be caused by the relative movement between the receiving end and the transmitting end or the movement of objects in the channel. The change of the channel with time, that is, the time-varying characteristics of the channel, is called the Doppler effect. If the receiver moves away from the transmitter at a speed $v$, the distance $l$ between the two will continue to increase. Then:

$$E(t) = E_0 \cdot \cos \left( 2\pi t \left( f_c - \frac{v}{\lambda} \right) - k_0 l_0 \right) \tag{13}$$

If the transmitter movement direction is consistent with the wave propagation direction, the Doppler effect is:

$$v = -\frac{v}{\lambda} = -f_c \cdot \frac{v}{c_0} \tag{14}$$

Otherwise, the Doppler effect is:

$$v = -\frac{v}{\lambda} \cos(\gamma) = -f_c \cdot \frac{v}{c_0} \cos(\gamma) \tag{15}$$

where $v \cos(\gamma)$ is the velocity component of the wave propagation direction.

In mobile communications, the Doppler effect is the main parameter that distinguishes other communication modes. Fading and Doppler shift cause a large number of consecutive bits to be destroyed. This leads to a significant drop in system performance, resulting in time-selective fading of the channel.

## 4. Contribution

There are a large number of scatterers such as buildings and trees in a complex urban environment. These scatterers have different distribution densities in different positions. And as the height changes, the density of the scatterers decreases. In emergency management, when the UAV interacts with the ground, the performance of the propagation channel plays an important role in the reliability and stability of communication. According to the different channel characteristics of UAVs in different regions, this section proposes to divide the channel into ultra-low altitude (0~100 m) and low altitude (100~1000 m) for modeling.

### 4.1. Construction of SUUL Model for Urban Environment Ultra-Low Altitude

At ultra-low altitudes, the distribution density of complex scatterers such as buildings and vegetation is relatively high. These scatterers cause various interferences such as reflection, scattering, and diffraction in the communication process of the UAV. In this case, the channel performance is mainly affected by shadow fading and multipath fading. In this section, lognormal shadow model is adopted to describe the path loss during signal transmission, and Rice distribution is used to describe the multipath fading, so as to establish the suitable channel model.

### 4.1.1. Analysis of the Ultra-Low Altitude Received Signal Envelope

When the UAV collects and transmits data in the ultra-low altitude environment, the received signal consists of LOS component and multipath component. Assume that the shadow only affects the LOS component. In the urban environment, the shadow fading is obvious. Considering the influence of the shadow in the communication channel, it is assumed that the LOS component obeys logarithmically distributed. The received signal is:

$$R(t) = W(t)e^{jF(t)} + A(t)e^{jF_0} \tag{16}$$

where $W(t)$ is the amplitude of the scatter component, which obeys the Rayleigh distribution, $F(t)$ is the phase of the multipath component, which is uniformly distributed in the interval $[-\pi, \pi)$. $A(t)$ is the amplitude of the LOS component, which obeys the Lognormal distribution, $F_0$ is the deterministic phase of the LOS component. $W(t)$, $F(t)$, $A(t)$ are independent of each other.

Assuming that $A(t)$ remains unchanged, the conditional probability density function of received signal envelope $R(t) = |R(t)|$, which obeys the Rice distribution is:

$$f_{R|A}(r|a) = \frac{r}{\lambda_0^2} \exp\left(-\frac{r^2 + a^2}{2\lambda_0^2}\right) I_0\left(\frac{ar}{\lambda_0^2}\right) \tag{17}$$

where $2\lambda_0^2$ is the average scattered power, and it satisfies $2\lambda_0^2 = E[W^2]$.

When $\alpha$ tend to 0, the power of the received signal consists of multipath component power, and there is no LOS component. Set $x = \frac{ar}{\lambda_0^2}$, $v = 1$:

$$I_0\left(\frac{ar}{\lambda_0^2}\right) \approx \frac{1}{\Gamma(1)} = 1 \tag{18}$$

Substitute Equation (18) into Equation (17):

$$f_{R|A}(r|a) = \frac{r}{\lambda_0^2} \exp\left(-\frac{r^2 + a^2}{2\lambda_0^2}\right) \tag{19}$$

At this time, $W$ obeys the Rayleigh distribution:

$$f_W(w) = \frac{w}{\lambda_0^2} \exp\left(-\frac{w^2}{2\lambda_0^2}\right) \quad , \qquad w \geq 0 \tag{20}$$

The conditional mathematical expectation of the received signal envelope is calculated by the total probability equation $f_R(r) = E_A[f_{R|A}(r|a)]$:

$$f_R(r) = \frac{r}{\lambda_0^2} \int_0^\infty \exp\left(-\frac{r^2 + a^2}{2\lambda_0^2}\right) I_0\left(\frac{ar}{\lambda_0^2}\right) f_A(a)\,\mathrm{d}a, \tag{21}$$

$$f_A(a) = \frac{1}{a\sqrt{2\pi d_0}} \exp\left(-\frac{(\ln a - \mu)^2}{2d_0}\right) \qquad (22)$$

where $\mu = E[\ln(A)]$, $d_0 = Var[\ln(A)]$.

The probability density function of the envelope of the received signal affected by shadow fading suitable for the Loo model is calculated:

$$f_R(r) = \frac{r}{\lambda_0^2 \sqrt{2\pi d_0}} \int_0^\infty \frac{1}{a} \exp\left(-\frac{\lambda_0^2(\ln a - \mu)^2 + d_0(r^2 + a^2)}{2\lambda_0^2 d_0}\right) I_0\left(\frac{ar}{\lambda_0^2}\right) da \qquad (23)$$

The channel conditions are estimated in real time through the data collected by the UAV. The parameters in Equation (21) are associated with the actual parameters $(d, \sigma_X^2, K_r)$ in order to calculate the shadow variance $\sigma_X^2$ [17] and the Rice factor $K$ [18]. Firstly, $Kr$ is derived from the probability rate density function of $K$. The calculated probability density function of K in the channel is as follows:

$$f_K(k) = \frac{1}{2k\sqrt{2\pi d_0}} \exp\left(-\frac{\left(\ln \lambda_0\sqrt{2k} - \mu\right)^2}{2d_0}\right) \qquad (24)$$

The Equation (24) is a probability density function of a lognormal variable. So:

$$\mu_{\ln k} = \ln K_r - 2d_0 \qquad (25)$$

$$Var[\ln(k)] = 4d_0 \qquad (26)$$

According to the first moment relation of the probability density function of the lognormal, the expression for $K_r$ can be obtained using Equation (27):

$$K_r = \frac{\mu_{R_a}}{2\lambda_0^2} \qquad (27)$$

where $\mu_{R_a}$ is the average power of the LOS component and also the second moment of the LOS component.

When both Rayleigh random processes and lognormal random processes are expressed in dB, they can be added and subtracted directly when calculating the gain or attenuation. Then the average value of received power is $\overline{S} = \mu_{R_a} + 2\lambda_0^2$, which can be calculated according to Equation (27). The respective terms are given by the expressions:

$$\mu_{R_a} = \frac{K_r}{1 + K_r}\overline{S} \qquad (28)$$

$$2\lambda_0^2 = \frac{\overline{S}}{K_r + 1} \qquad (29)$$

Substituting the above equations into Equation (22):

$$2\lambda_0^2 = \frac{1}{K_r + 1} P_t \left(\frac{\lambda}{4\pi d_{ref}}\right)^2 \left(\frac{d_{ref}}{d}\right)^\alpha \qquad (30)$$

The second moment [19] of the LOS component is denoted by $\mu_{R_a} = \exp(2\mu + 2d_0)$:

$$\mu = \frac{1}{2}\ln\left(\frac{K_r}{1 + K_r} P_t \left(\frac{\lambda}{4\pi d_{ref}}\right)^2 \left(\frac{d_{ref}}{d}\right)^\alpha\right) - d_0 \qquad (31)$$

We set $\Psi = 10\lg(P_t/R_a)$, where $R_a$ is the received LOS component power, and the variance of $\Psi$ is equal to the shadow variance [20]. Then the probability density function of $\Psi$ is:

$$f_\Psi(\Psi) = \frac{1}{2\zeta\sqrt{2\pi d_0}}\exp\left(-\frac{(\Psi - (10\lg P_t - 2\zeta\mu))^2}{8\zeta^2 d_0}\right) \tag{32}$$

where $\Psi$ obeys normal distribution, $\zeta = 10/\ln(10)$. Then $d_0$ can be expressed as:

$$d_0 = \frac{\sigma_X^2}{4\zeta^2} \tag{33}$$

According to the Equation (23) we associate the model with the estimated parameters, and obtain the probability function of the received signal power as:

$$f_S(s) = \frac{1}{2\lambda_0^2\sqrt{2\pi d_0}}\int_0^\infty \frac{1}{a}\exp\left(-\frac{\lambda_0^2(\ln a - \mu)^2 + d_0(s + a^2)}{2\lambda_0^2 d_0}\right)I_0\left(\frac{a\sqrt{s}}{\lambda_0^2}\right)da \tag{34}$$

where $S$ is the power of the received signal.

### 4.1.2. BER Analysis of SUUL Model

According to the intermediate variable $\gamma = \frac{sT_s}{N_0}, \overline{\gamma} = \frac{T_s}{N_0}\overline{s}$, so:

$$f_\gamma(\gamma) = \frac{N_0}{T_s}f_S(s) = \frac{N_0}{T_s}f_S\left(\frac{N_0}{T_s}\gamma\right) = \frac{\overline{s}}{\overline{\gamma}}f_S\left(\frac{\overline{s}}{\overline{\gamma}}\gamma\right) \tag{35}$$

The probability density function of signal-to-noise ratio (SNR) is:

$$f_\gamma(\gamma) = \frac{\overline{s}}{2\overline{\gamma}\lambda_0^2\sqrt{2\pi d_0}}\int_0^\infty \frac{1}{a}\exp\left(-\frac{\lambda_0^2(\ln a - \mu)^2 + d_0\left(\frac{\overline{s}}{\overline{\gamma}}\gamma + a^2\right)}{2d_0\lambda_0^2}\right)I_0\left(\frac{a}{\lambda_0^2}\sqrt{\frac{\overline{s}}{\overline{\gamma}}\gamma}\right)da \tag{36}$$

where $\overline{\gamma}$ is the average signal-noise ratio of per symbol, $\sigma_X^2$ is the shadow variance.

According to the following equation [21] the expression of bit error rate (BER) can be obtained:

$$P_b(E) = \int_0^\infty P_b(E; \gamma)f_\gamma(\gamma)d\gamma \tag{37}$$

For PSK modulation, the expression of BER is calculated as follows:

$$\xi = \frac{\overline{s}}{2\lambda_0^2\gamma\sqrt{2\pi d_0}}\int_0^\infty Q\left(\sqrt{2\gamma}\right)\int_0^\infty \frac{1}{a}\exp\left(-\frac{\lambda_0^2\overline{\gamma}(\ln a - \mu)^2 + d_0(\overline{s}\gamma + \overline{\gamma}a^2)}{2d_0\lambda_0^2\overline{\gamma}}\right)I_0\left(\frac{a}{\lambda_0^2}\sqrt{\frac{\overline{s}\gamma}{\overline{\gamma}}}\right)da\,d\gamma \tag{38}$$

The $\xi$ is normalized so that it is only related to the average signal-to-noise ratio per bit. The following equations are used for normalization:

$$a_n = \frac{a}{\sqrt{\overline{s}}} \tag{39}$$

$$2\lambda_{0n}^2 = \frac{1}{K_r + 1} \tag{40}$$

$$\mu_n = \frac{1}{2}\ln\left(\frac{K_r}{1 + K_r}\right) - d_0 \tag{41}$$

The normalized BER is expressed as:

$$\xi = \frac{1}{2\lambda_{0n}^2\gamma\sqrt{2\pi d_0}}\int_0^\infty Q(\sqrt{2\gamma})\int_0^\infty \frac{1}{a_n}\exp\left(-\frac{\lambda_{0n}^2\overline{\gamma}(\ln a_n - \mu_n)^2 + d_0(\gamma + a_n^2)}{2d_0\lambda_{0n}^2\overline{\gamma}}\right)I_0\left(\frac{a_n}{\lambda_{0n}^2}\sqrt{\frac{\gamma}{\overline{\gamma}}}\right)da_n\,d\gamma \tag{42}$$

However, Equation (42) cannot be implemented well in simulation. The model provided in [22] continues to be used as an approximation to derive the BER, taking into account computational complexity and other factors. This model discusses the same channel characteristics as the model in this article.

The Nakagami model is proposed in [23], and the magnitude of the LOS component obeys the Nakagami distribution:

$$f_A(a) = \frac{2m^m a^{2m-1}}{\mu_{R_a}{}^m \Gamma(m)} \exp\left(-\frac{ma^2}{\mu_{R_a}}\right) \quad , \qquad a \geq 0 \tag{43}$$

where $\Gamma(\cdot)$ is the Gamma function and $m$ is the parameter of the Nakagami distribution, $m = \frac{\mu_{R_a}{}^2}{Var[R_a]} \geq 0$.

In order to better model various LOS conditions in the channel, let the range of $m$ be $[0, \infty)$. In a multipath fading environment [21] the distribution parameter of the traditional Nakagami model ranges from $[0.5, \infty)$. A suitable $m$ value is chosen to simulate LOS conditions in urban areas.

The model parameter $(\mu_{R_a}, m, \lambda_0^2)$ are associated with the measurement parameter $(d, K_r, \sigma_X^2)$ in order to better apply the approximate model. Derive the approximate probability density function of the variance of $\Psi$:

$$f_\Psi(\Psi) = \frac{1}{\Gamma(m)\zeta} \left(P_t \frac{m}{\mu_{R_a}}\right)^m \exp\left(-\frac{m}{\zeta}\Psi - \frac{mP_t}{\mu_{R_a}\zeta} \exp\left(-\frac{\Psi}{\zeta}\right)\right) \tag{44}$$

The parameters are correlated with those in the LOO model and use the second-order moment [24] to get the expressions of the following parameters:

$$d_0 = \frac{1}{4}\Psi'(m) \tag{45}$$

$$\mu_{R_a} = m\exp(2\mu - \Psi(m)) \tag{46}$$

where $\Psi'(\cdot)$ is the first partial derivative of $\Psi(\cdot)$ [21].

$\mu_{R_a}$ and $m$ can be calculated according to Equations (46) and (45). The probability density function of the received signal envelope can be calculated by Equations (17) and (43) and conditional mathematical expectation:

$$f_R(r) = \left(\frac{2\lambda_0^2 m}{2\lambda_0^2 m + \mu_{R_a}}\right)^m \frac{r}{\lambda_0^2} \exp\left(-\frac{r^2}{2\lambda_0^2}\right) {}_1F_1\left(m; 1; \frac{\mu_{R_a}r^2}{2\lambda_0^2(2\lambda_0^2 m + \mu_{R_a})}\right) \quad , \ r \geq 0 \tag{47}$$

where ${}_1F_1(\ ; \ ; \ )$ is the confluent hypergeometric function [21].

Thus, the probability density function of the received power is:

$$f_S(s) = \left(\frac{2\lambda_0^2 m}{2\lambda_0^2 m + \mu_{R_a}}\right)^m \frac{1}{2\lambda_0^2} \exp\left(-\frac{s}{2\lambda_0^2}\right) {}_1F_1\left(m; 1; \frac{\mu_{R_a}s}{2\lambda_0^2(2\lambda_0^2 m + \mu_{R_a})}\right) \quad , \ s \geq 0 \tag{48}$$

Since $f_\gamma(\gamma) = \frac{\bar{s}}{\bar{\gamma}}f_S\left(\frac{\bar{s}}{\bar{\gamma}}\gamma\right)$, the probability density function of $\gamma$ can be expressed as:

$$f_\gamma(\gamma) = \left(\frac{2\lambda_0^2 m}{2\lambda_0^2 m + \mu_{R_a}}\right)^m \frac{\bar{s}}{2\lambda_0^2\bar{\gamma}} \exp\left(-\frac{\bar{s}\gamma}{2\lambda_0^2\bar{\gamma}}\right) {}_1F_1\left(m, 1, \frac{\mu_{R_a}\bar{s}\gamma}{2\lambda_0^2\bar{\gamma}(2\lambda_0^2 m + \mu_{R_a})}\right) \tag{49}$$

The moment generating function can be expressed as [21]:

$$M_\gamma(s) = \frac{\left(2\lambda_0^2 m\right)^m \left(1 + \frac{2\lambda_0^2}{\bar{s}}\bar{\gamma}s\right)^{m-1}}{\left(\left(2\lambda_0^2 m + \mu_{R_a}\right)\left(1 + \frac{2\lambda_0^2}{\bar{s}}\bar{\gamma}s\right) - \mu_{R_a}\right)^m} \tag{50}$$

Calculate the BER in the ideal coherent BPSK signals as given by Equation (51):

$$\xi = \frac{1}{\pi}\int_0^{\frac{\pi}{2}} M_\gamma\left(\frac{1}{\sin^2\theta}\right) \mathrm{d}\theta = \frac{\left(2\lambda_0^2 m\right)^m \left(1 + \frac{2\lambda_0^2}{\bar{s}}\bar{\gamma}\right)^{m-1}}{4\left(2\lambda_0^2 m + \frac{2\lambda_0^2}{\bar{s}}\bar{\gamma}\left(2\lambda_0^2 m + \mu_{R_a}\right)\right)^m} \times$$

$$\quad {}_1F_1\left(\frac{1}{2}, 1-m, m; 2; \frac{1}{\left(1+\frac{2\lambda_0^2}{\bar{s}}\bar{\gamma}\right)}, \frac{2\lambda_0^2 m}{\left(2\lambda_0^2 m + \frac{2\lambda_0^2}{\bar{s}}\bar{\gamma}\left(2\lambda_0^2 m + \mu_{R_a}\right)\right)}\right) \tag{51}$$

According to $\lambda_{0n}^2 = \frac{\lambda_0^2}{\bar{s}}$, $\mu_{R_a n} = \frac{\mu_{R_a}}{\bar{s}}$ the normalized BER can be obtained.

$$\xi = \frac{\left(2\lambda_{0n}^2 m\right)^m \left(1+2\lambda_{0n}^2\bar{\gamma}\right)^{m-1}}{4\left(2\lambda_{0n}^2 m + 2\lambda_{0n}^2\bar{\gamma}\left(2\lambda_{0n}^2 m + \mu_{R_a n}\right)\right)^m}$$

$$\quad {}_1F_1\left(\frac{1}{2}, 1-m, m; 2; \frac{1}{\left(1+2\lambda_{0n}^2\bar{\gamma}\right)}, \frac{2\lambda_{0n}^2 m}{\left(2\lambda_{0n}^2 m + 2\lambda_{0n}^2\bar{\gamma}\left(2\lambda_{0n}^2 m + \mu_{R_a n}\right)\right)}\right) \tag{52}$$

### 4.2. Construction of Low-Altitude SULA Model of Urban Environment

In the low-altitude environment, the distribution density of complex scatterers decreases as the flying height of the UAV increases. The influence of shadow fading on the channel becomes smaller. At this time, the Doppler effect caused by the relative movement of the UAV and the ground base station plays a major role in the signal fading. The rate of change of time-varying signal envelope is proportional to Doppler spread spectrum. Large Doppler shift causes rapid changes in the time-varying signal envelope and distorts the signal. In the small-scale flat fading channel described by the Clarke channel model, the received signals of mobile stations have the statistical characteristics of the field intensity based on the scattering mode. This is in line with the characteristics of low-altitude urban mobile communication environment where there is no direct path. The Jakes' model fits well with the statistical characteristics of the Clarke model, so it is optimized based on the Jakes' model and the beamforming technology is adopted to achieve the compensation of the Doppler frequency shift to improve the communication reliability.

#### 4.2.1. Construction of Beamforming in Low Altitude Environment

In the process of signal transmission, the different path signals generated by the multipath effect reach the receiving end at different angles. The different paths of the signal are distinguished according to the arrival angle of the wave. Referring to Jakes' model, it is assumed that the path generated by the multipath channel can be completely separated. Signals arriving from different paths have their own characteristics. The baseband time-varying multipath channel is:

$$h(n,\eta) = \sum_{u=1}^{u} h_u(n)\sigma(\eta - \eta_u) \tag{53}$$

where $\eta_u$ is the sampling delay, $u$ is the path of signal transmission, $h_u(n)$ is a different sub-path. The signals transmitted by different paths will experience different attenuation due to the distribution of obstructions. When they reach the receiving end from different angles, the signals on each path will produce different Doppler shifts:

$$h_u(n) = \bar{h}_u \exp(j2\pi nTf_D\cos\theta_u + j\psi_u) \tag{54}$$

where $f_D$ is the maximum Doppler frequency, $\bar{h}_u$ is the attenuation of the signal, $\theta_u$ is the angle of arrival of the signal, $\psi_u$ is the phase shift of the $u$th path, subject to uniform distribution, and $T$ is the sampling interval.

The omnidirectional antenna has better performance in high mobility scenarios. Therefore, Uniform circular array (UCA) is adopted to achieve beamforming in order to cover the received signal in all directions. As shown in Figure 4. The spacing of each element is $l$, and the first element is taken as a reference. The received signal in the time domain at the $i$th element is:

$$s_i(n) = \sum_{u=1}^{u} \bar{h}_u x(n - \eta_u) \exp\left( j2\pi \left( \frac{R}{\lambda} \cos(\theta_u - \alpha_i) + nT f_D \cos\theta_u \right) \right) + z_i(n) \tag{55}$$

where $R = Il/2\pi$ is the radius of the uniform circular array, $\lambda$ is wavelength of the carrier, $\alpha_i = (i-1)\frac{2\pi}{T}$ is the polar angle of the $i$th element, and according to theoretical calculations $R$ is about 30 cm. $Z_i = \left[ Z_i(-N_g), \cdots, z_i(N-1) \right]^T$ is the additive white Gaussian noise.

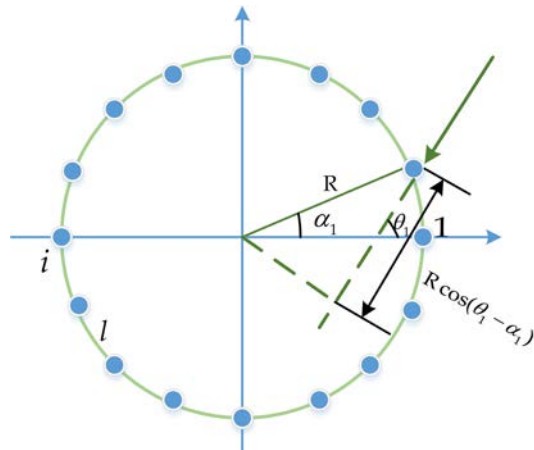

**Figure 4.** Uniform circular array.

When the UAV transmits data, the signal on each path will produce a different Doppler frequency shift. The Doppler shift can be estimated if it is assumed that there is only one angle of arrival.

Firstly, the uniform circular array beamforming technology is used to separate the Doppler frequency shift caused by the multipath effect. A beamforming network with $u$ different paths contains $u$ beamforming vectors such as $\gamma_i = [\gamma^{(1)}, \cdots, \gamma^{(u)}]$.

The $\gamma^{(u)}$ of each $u$ is an $I \times 1$ vector. The received signal after beamforming is:

$$s(u) = \gamma^{(u)H}s, 1 \le u \le U \tag{56}$$

where $s$ is the receiving signal stack on the antenna, which meets the condition:

$$s_i = [s_i(-N_g), \cdots, s_i(N-1)]^T \tag{57}$$

Assuming that the received signal meets the far-field condition. Equation (58) is used to represent the steering vector in $\theta$ direction of a narrowband signal with carrier frequency $f_c$:

$$V(\theta) = \left[ e^{j2\pi R \cos(\theta - \alpha_1)/\lambda}, \cdots, e^{j2\pi R \cos(\theta - \alpha_I)/\lambda} \right]^T \tag{58}$$

The least square method is used to calculate the weight $\gamma^{(u)}$. Equation (59) can be obtained by the radiation pattern constraints on the path direction vector:

$$R = [V(\theta_1), V(\theta_2), \cdots, V(\theta_u)] \tag{59}$$

The expected output under the radiation pattern constraints is represented by $G_R = R^H \vec{\gamma}$. At the same time, the least squares constraint is set in $Q$ directions, and then:

$$Z = \left[ V(\theta_{z_1}), V(\theta_{z_2}), \cdots, V\left(\theta_{z_Q}\right) \right] \tag{60}$$

The expected output at this time is $G_Z = Z^H \vec{\gamma}$.

The cost function is established by taking $E = G_Z - Z^H \vec{\gamma}$ as the error vector is shown in Equation (61):

$$F\left(\vec{\gamma}\right) = E^H E = \left( G_Z - Z^H \vec{\gamma} \right)^H \left( G_Z - Z^H \vec{\gamma} \right) \tag{61}$$

According to Equation (61) and the expected output the beamforming weights can be expressed as:

$$\vec{\gamma} = \left( ZZ^H \right)^{-1} \left( ZG_Z - R\left( R^H \left( ZZ^H \right)^{-1} R \right)^{-1} \cdot \left( R^H \left( ZZ^H \right)^{-1} ZG_Z - G_R \right) \right) \tag{62}$$

Set $G_Z = 0$:

$$\vec{\gamma}^{(u)} = \left( ZZ^H \right)^{-1} R\left( R^H \left( ZZ^H \right)^{-1} R \right)^{-1} G_R^{(u)}, 1 \le u \le U \tag{63}$$

### 4.2.2. Design of Doppler Frequency Shift Compensation Algorithm

The influence of Doppler shift on the performance of UAV communication channel cannot be ignored. It causes a large number of continuous bits to be destroyed, data packet loss, and signal distortion problems. Assuming that the signal is only affected by the multipath effect and there is no Doppler effect, the received signal is:

$$s(t) = H(\omega)e^{j\omega t} \tag{64}$$

where $H(\omega)$ is:

$$H(\omega) = \sum_{u=1}^{U} a_u e^{-j\omega \tau_u} \tag{65}$$

When the multipath effect and Doppler effect exist at the same time, suppose the Doppler angle frequency shift of the $u$th ray is $\omega_u = 2\pi f_u$, and $|f_u| \le f_D$. Then the received signal is:

$$s(t) = H(\omega, t)e^{j\omega t} \tag{66}$$

where $H(\omega)$ is:

$$H(\omega, t) = \sum_{u=1}^{U} a_u e^{-j\omega \tau_u + j\omega_u t} \tag{67}$$

where $U$ is the total number of arrival paths, $a_u$ is the amplitude of the $u$th ray, $\tau_u$ is the arrival time.

Assuming that the mobile communication transmits a time harmonic signal of $f = 10$ Hz, and there are 16 arrival paths. The experimental results are shown in the Figure 5 for the presence and absence of Doppler effect.

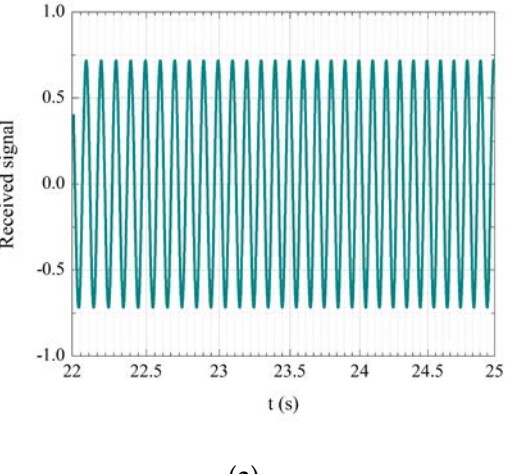 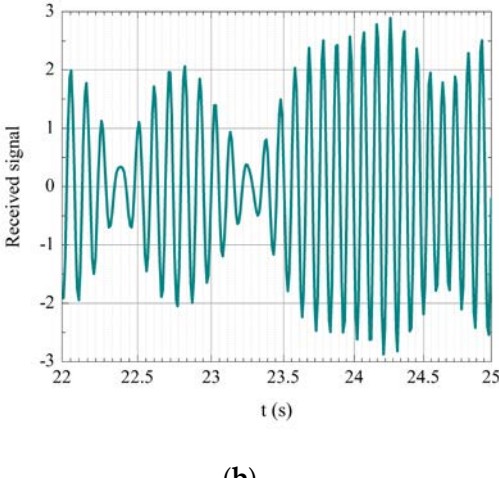

(**a**)  (**b**)

**Figure 5.** The multipath effect of the signal and the combined effect of multipath and Doppler of the signal. (**a**) No Doppler frequency shift; (**b**) With Doppler frequency shift.

When there is no Doppler frequency shift, the received signal is still a harmonic signal with time $f$ = 10 Hz, and there is no distortion. When there is a Doppler frequency shift, the received signal is distorted and changes continuously with the increase of time, resulting in a decrease in communication reliability. The results show that the frequency shift caused by the Doppler effect degrades the BER performance of UAV communication. In order to meet the communication reliability required by emergency management, it is necessary to correct the data by compensating the Doppler frequency shift.

The Doppler frequency is separated based on the AOAs signal, and the Doppler frequency of different paths is compensated. When the maximum Doppler frequency $f_D$ and the arrival angle $\theta_u$ of the path $u$ are obtained, the compensation frequency of the path is $f_D \cos \theta_u$.

The Doppler frequency shift compensation of path $u$ is:

$$\tilde{s}^{(u)}(n) = e^{-j2\pi f_D nT \cos \theta_u} s^{(u)}(n) \tag{68}$$

Without linear interpolation and fast Fourier transform, the received signal is:

$$S^{(u)}(k) = \frac{1}{\sqrt{n}} \sum_{n=1}^{N-1} \tilde{s}^{(u)}(n + N_g) e^{-j2\pi nk/N} \tag{69}$$

## 5. Simulation Results

### 5.1. Simulation of SUUL Model

The Minimum Shift Keying (MSK) modulation method is adopted in the traditional Loo model. However, there are a lot of obstacles in the communication of ultra-low altitude UAVs. The fading and shadow in the channel will cause the phase change of the signal. The traditional coherent demodulation method of MSK is not suitable for this situation. Therefore, the binary phase shift keying (BPSK) modulation method is adopted in this paper. In the simulation experiment, the influence of three physical parameters in the UAV channel is considered: the average signal-to-noise ratio per unit bit, the shadowing standard deviation, and the average Rice factor. In addition, BER is an important evaluation index in the evaluation of channel performance. Especially in the field of emergency management, urgent communication process, link interruption and data error will have a great impact on the whole system. As a result, the BER performance of the three channel models were compared under the same parameter conditions.

The relationship between BER and average signal-to-noise ratio per bit is shown in Figure 6. The three curves in the figure are: the BER curve of the Loo model in MSK mode, ideal channel curve and SUUL channel curve.

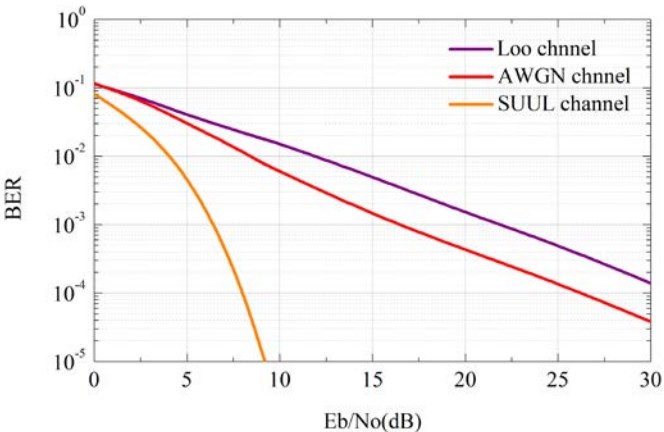

**Figure 6.** Curve of relationship between BER and average signal-noise ratio per bit.

When the SNR is approximately 10 dB, the BER of SUUL model is $10^{-5}$. At this point, the BER of LOO model is $10^{-2}$. It can be seen that in the same communication environment, the bit error rate corresponding to the reliability of UAVs in emergency communications of the SUUL model is much lower than that of the LOO model. High reliability means that the communication link is more stable and less prone to packet loss and data distortion, so the BER is essential for emergency communications.

The relationship between bit error rate and shadow standard deviation is shown in Figure 7. $K_r$ is the Rice factor, which is defined as the ratio of the power of the main signal to the multipath component. The $K_r$ is used to reflect the influence of small-scale fading. When $K_r$ changes from 0 dB to 20 dB, it can be seen that the overall change trend of the channel error rate is similar. This means that the SUUL channel has good stability in different small-scale fading environments.

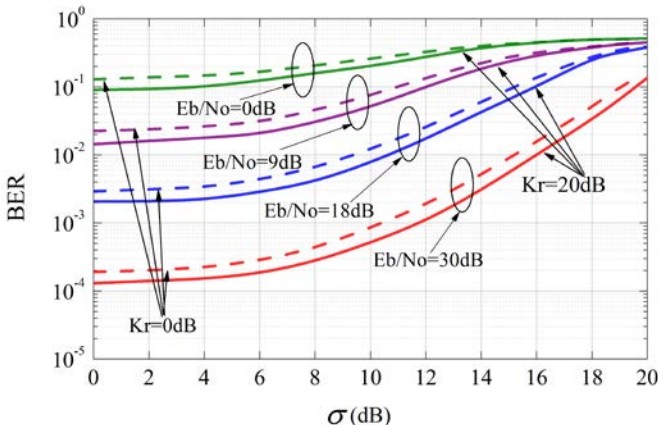

**Figure 7.** Relationship between BER and shadowing standard deviation.

On the other hand, the variation range of the BER curve is quite different under different SNR scenarios. When the SNR is small, the bit error rate changes relatively smoothly with the increase of the shadow standard deviation. At this time, the shadow has little effect on the UAV channel performance. When the SNR reaches a medium or high value, the change in the bit error rate is obvious, which means that the shadow has a greater impact on the performance of the communication channel. In other words, when the SNR is large, the channel performance is affected by both shadow and small-scale fading. When

the SNR of the received signal is small, corresponding to long-distance communication, the influence of the shadow on the channel can be approximately ignored at this time.

The relationship between the uncertainty of SUUL model and shadow standard deviation is shown in Figure 8. Uncertainty refers to the difference of BER when $K_r$ is 0 dB and 20 dB under the same condition of signal-to-noise ratio. It is the influence of small-scale fading on the received signal under the same shadow fading environment and the same signal-to-noise ratio. It can be seen that the system uncertainty decreases non-linearly as the shadowing standard deviation increases. In the near field communication environment, when the signal-to-noise ratio reaches a larger value of about 18 dB to 30 dB, the system uncertainty value is very large. At this time, the shadow fading has a major influence on the UAV channel. In long-distance communication, that is, when the signal-noise ratio is 0 dB, the system uncertainty value is the smallest, and the influence of shadow on BER can be ignored, but the influence of small-scale fading needs to be considered. Therefore, it is not difficult to find that the channel characteristics change as the communication distance changes. That is, when the altitude changes from ultra-low altitude to low altitude, different channel models should be established for the change of channel characteristics to improve channel performance.

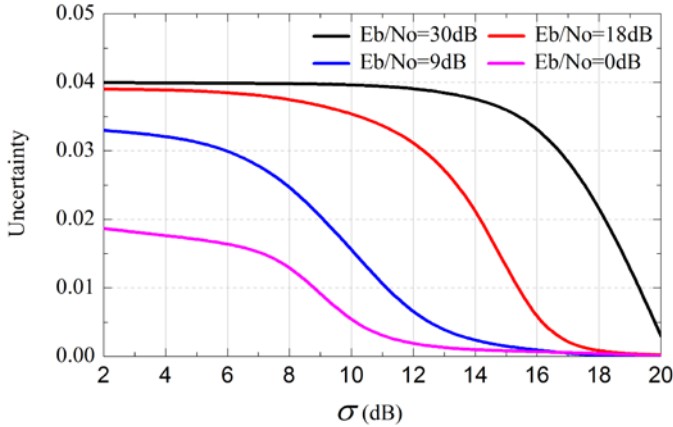

**Figure 8.** The relationship between uncertainty and shadowing standard deviation.

### 5.2. Simulation of the SULA Model

The Doppler effect is caused by relative motion. Figure 9a is the influence of Doppler effect on the received signal envelope when the speed is 80 km/h in the Rayleigh channel. When the speed is 120 km/h, the influence of Doppler effect is shown in Figure 9b.

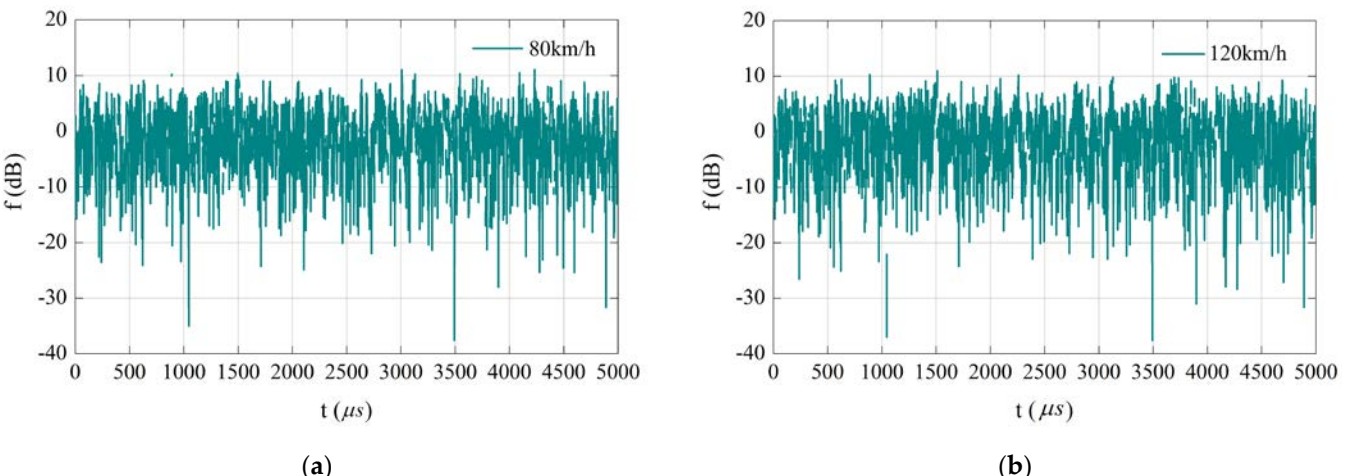

(**a**)  (**b**)

**Figure 9.** Influence of Doppler on signal fading at different velocity. (**a**) Influence of Doppler effect on channel fading at 80 km/h flight speed; (**b**) Influence of Doppler effect on channel fading at 120 km/h flight speed.

According to the simulation results, the maximum envelope attenuation of the UAV at high speed is greater than that at low speed. It can be seen that the faster the UAV goes, the stronger the signal attenuation is and vice versa. As the speed increases, the influence of the Doppler effect on the envelope of the received signal becomes more and more obvious. In the process of information transmission, a large number of consecutive bits are destroyed due to fading and Doppler frequency shift, which leads to an increase in the bit error rate.

Figure 10 shows the BER performance of a single antenna receiver, a 16-element diversity receiver, and a 16-element beamforming receiver under three different path numbers.

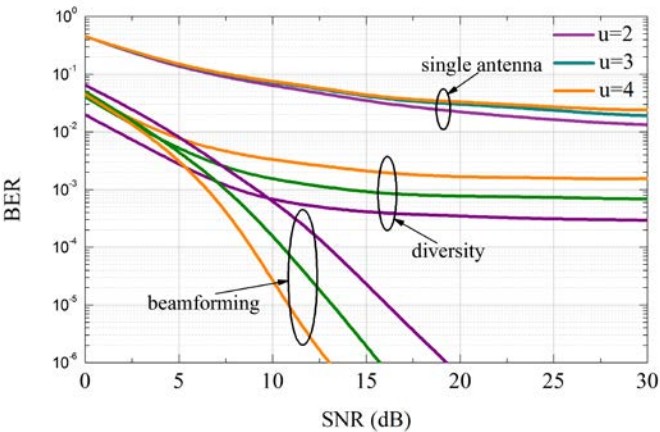

**Figure 10.** The relationship between the BER and the SNR of the three methods.

It can be seen from Figure 10 that the bit error rate of a single antenna channel is significantly higher than the other two methods under the same channel conditions. When the signal-to-noise ratio is small, the channel error rate of the multi-antenna diversity method is similar to the SULA channel established in this paper. However, because the multi-antenna diversity technology does not separate the directional path of each unit, the influence of Doppler frequency shift on the path still exists, which will lead to frequency domain interference and increase in bit error rate. Therefore, with the increase of the signal-to-noise ratio, the bit error rate of the SULA channel decreases significantly, but the bit error rate of the multi-antenna diversity channel changes slightly. In contrast, the SULA model uses uniform circular array beamforming technology to separate the Doppler frequencies of different arrival paths and then correct them separately, which can better eliminate the influence of Doppler frequency shift on the channel. In particular, in Figure 10, the orange line represents the system bit error rate curve when the number of paths is larger. The bit error rate of channels using single-antenna and multi-antenna diversity technologies increases with the increase in the number of paths. However, the channel model proposed in this paper first separates the paths, and then compensates for the Doppler shift. As the number of paths increases, the algorithm gain effect becomes more significant, and the system error rate is still decreasing.

The BER performance of different normalized Doppler frequencies in the three cases is shown in Figure 11. The number of $u$ is fixed at 3, the antenna unit is 16, and the SNR is 15 dB. It can be seen that the performance of multi-antenna diversity method and beamforming method is much better than that of single antenna method. However, with the increase of the normalized Doppler frequency, the bit error rate of the multi-antenna diversity method increases sharply. Correspondingly, the performance of the SULA model remains stable under different Doppler extensions. Therefore, when the UAV is carrying out data transmission, even if the communication environment is constantly changing and becoming more complex, the bit error rate of this model is still kept in a small range. This ensures the reliability of communication when emergency work is in progress.

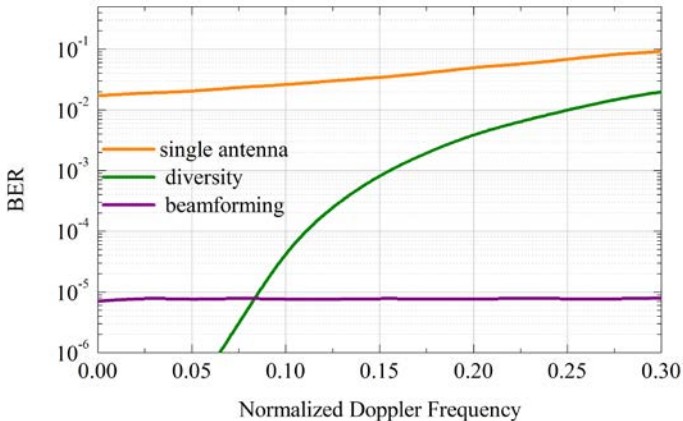

**Figure 11.** Relationship between BER and normalized Doppler frequency.

## 6. Discussion

On the one hand, all research in this paper is based on the analysis of the channel characteristics of the urban environment. However, in the suburban environment, the density of buildings and vegetation is low, and the LOS component is more likely to exist. It is different from the channel characteristics of the city. In order to make the model universal and have better performance in the suburban environment, the model needs to be further improved and updated. Such channel models for different scenarios need to be further studied. On the other hand, the method of adaptive adjustment of ultra-low altitude and low-altitude channel models needs subsequent improvement. This article determines the channel selection based on the intuitive flying height of the UAV. However, there are differences in the layout of buildings in different countries and cities. If the channel parameters measured by UAVs are used as adaptive standards, whether the channel model reliability will be further improved remains to be studied. For example, when the channel parameters reach a certain value, the channel model is automatically adjusted. This will be the main direction of the follow-up research.

## 7. Conclusions

In an urban environment, the distribution density and position of complex scatterers at different heights are different. This causes differences in channel characteristics. In the face of the complex and changeable channel characteristics of the urban environment, the errors caused by using a single channel have brought huge challenges to the reliability of data transmission. In the urban emergency management, in order to reduce the attenuation, interference, loss, distortion and packet loss of data during the transmission process, a more accurate adaptive SUUL-SULA channel model is established in this paper. The channel is divided into two parts according to height: ultra-low altitude (0~100 m) and low altitude (100~1000 m). First of all, in the ultra-low altitude, the distribution density of obstructions such as buildings and trees is relatively high. Due to the influence of these complex scatterers, large-scale fading and small-scale fading make the signal experience a lot of loss and attenuation on different transmission paths. Therefore, based on the LOO model, the SUUL model proposed in this paper assumes that the LOS component obeys the lognormal distribution, and the multipath component obeys the Rayleigh distribution. Under this condition, the probability density function and bit error rate of the received signal are derived. Secondly, in the low-altitude environment, as the UAV's flight altitude increases, the influence of shadow fading on the channel gradually weakens. At this time, the Doppler frequency shift caused by the flying speed of the UAV has a major impact on the signal attenuation. A SULA channel model is proposed based on the Jakes' model. The uniform circular array beamforming technology is adopted to design the Doppler frequency shift compensation algorithm. Finally, the experimental results show that the

SUUL-SULA channel has a good improvement in BER performance and system stability, which improves the reliability of the UAV communication.

**Author Contributions:** Methodology, software and writing—original draft preparation, Bing Han; formal analysis, Bing Han and Danyang Qin; writing—review and editing, Bing Han, Danyang Qin, Lin Ma, Ping Zheng and Merhawit Berhane Teklu; resources and project administration, Danyang Qin; supervision, Danyang Qin, Lin Ma, Ping Zheng and Merhawit Berhane Teklu All authors have read and agreed to the published version of the manuscript.

**Funding:** This research was funded by the National Natural Science Foundation of China (61771186), Outstanding Youth Project of Provincial Natural Science Foundation of China in 2020 (YQ2020F012) and Nursing Program for Young Scholars with Creative Talents in Heilongjiang Province (UNPYSCT-2017125).

**Conflicts of Interest:** The authors declare no conflict of interest.

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
