# Peer review of "Modeling and Performance Optimization of Unmanned Aerial Vehicle Channels in Urban Emergency Management"

_ijgi, doi:10.3390/ijgi10070478_

Round 1

Reviewer 1 Report

Good paper about application of UAV in urban emergency management 

Reviewer 2 Report

The manuscript entitled "Modeling and performance optimization of UAV channel in urban emergency management" presented the performance optimization in a promising way. However, it needs some points that need serious consideration.

  1. The paper needs serious editing in terms of English and scientific writing style as some parts of the paper are misleading or have no logical order or flow due to lengthy sentences. The paper must go through a rigorous editing and proofreading process before resubmission.
  2. The main limitation of the paper is that it lacks critical related work. The historical perspective should be discussed as well. The proposed study is not critically evaluated and compared to the related work/state-of-the-art and is not identified and discussed its drawbacks and limitations. Thus, it is not easy to assess the real contribution of the paper in the field and how much is efficient in the proposed study compared to related works. A clear assessment of the contribution of the authors when compared to existing approaches should be given. It is advised to add the "Literature Review" section after the Introduction.
  3. The figures need to be reproduced with at least 300dpi resolution as in the current resolution it's difficult to read and observe.
  4. It is advised to revise the Abstract and Conclusion. Both sections should be consistent in terms of Proposal, Problem statement, Results, and future work. As in the current format, both are opposing each other a bit.

Reviewer 3 Report

The manuscript addresses modeling and performance optimization of UAV channels in urban emergency management.

The paper does correspond to the structure of the paper to be submitted in this Journal. More delineation should be set between Intro, Related work, Methods, Results, and Conclusions. I also would like to see a separate Discussion of Results rather than stating them. Moreover, the Analysis of Models is too much in detail, but the proper discussion is lacking.

The quality of Figure 2 is suboptimal

Way too many formulas: I understand that authors try to infer the meaning and design of certain models better. However, from the readers’ point, these formulas are way too in detail and certainly will affect the readers' interest negatively. Moreover, the readers will lose track of all (unnecessary) formulas if they are not fed with some other content in between these formulas or if they are not visualized and connected to the cases and examples visually.  Unless the paper is being published in a math journal, the paper should be relatively visual. Thus, I would propose to decrease some of the formulas and exchange them with more (at least) schematic figures or visuals.

I would like to see more emphasis on the Discussion part. Discussion of Results should be better exposed and highlighted, rather than stating or listing their observed results.

Reviewer 4 Report

This is an interesting paper, relevant to the scope of the journal. I enjoyed reading it, and have some minor comments/suggestions for improvement:

  • It’s hard to stay focused on the methodology of the work with all these formulas and equations. My recommendation would be to reduce them if possible, and instead refer the readers to another source where the formula can be found
  • The related work section must be improved. There are several other works in the literature that are relevant to this study and should be discussed/cited. Please improve this.
  • The results and discussions section, especially with regards to the analyzed and arguments could be improved.
  • I highly recommend the authors to ask a native English speaker to proofread the work.

Round 2

Reviewer 2 Report

The manuscript is improved a lot as per suggestions provided and now it better presents the idea........Best of Luck